# Rationale and study protocol of the Physical Activity and Dietary intervention in women with OVArian cancer (PADOVA) study: a randomised controlled trial to evaluate effectiveness of a tailored exercise and dietary intervention on body composition, physical function and fatigue in women with ovarian cancer undergoing chemotherapy

Stephanie Stelten,[1,2] Meeke Hoedjes,[3] Gemma G Kenter,[4,5,6] Ellen Kampman,[7] Rosalie J Huijsmans,[8] Luc RCW van Lonkhuijzen,[5] LM Buffart 1,2

▶ Prepublication history and additional materials for this paper is available online. To view these files, please visit the journal online (http://dx.doi.org/10.1136/bmjopen-2020-036854).

For numbered affiliations see end of article.

**Correspondence to**
Dr LM Buffart;
laurien.buffart@radboudumc.nl

## ABSTRACT

**Introduction** As a consequence of ovarian cancer and its treatment, many women with ovarian cancer have to deal with reduced physical function, fatigue, and loss of weight and/or muscle mass, compromising quality of life. Exercise and dietary interventions can positively influence body composition, physical fitness and function, and fatigue in patients with cancer. However, there are no data from randomised controlled trials on the effectiveness of exercise and dietary interventions in patients with ovarian cancer. Due to a complex disease trajectory, a relatively poor survival and distinct disease-induced and treatment-induced side effects, it is unclear whether exercise and dietary interventions that were shown to be feasible and effective in other types of cancer produce comparable results in patients with ovarian cancer. The aim of this article is to present the design of the multicentre randomised controlled Physical Activity and Dietary intervention in OVArian cancer trial and to describe how the exercise and dietary intervention is tailored to specific comorbidities and disease-induced and treatment-induced adverse effects in patients with ovarian cancer.

**Methods and analysis** Adult women with primary epithelial ovarian cancer who are scheduled to undergo first-line (neo)adjuvant chemotherapy (n=122) are randomly allocated to a combined exercise and dietary intervention or a usual care control group during chemotherapy. Primary outcomes are body composition, physical function and fatigue. Outcome measures will be assessed before the start of chemotherapy, 3 weeks after completion of chemotherapy and 12 weeks later. The exercise and dietary intervention was tailored to specific

### Strengths and limitations of this study

► This is a randomised controlled trial in women with ovarian cancer.
► Systematic development and tailoring of the exercise and dietary intervention will improve compliance to the intervention and prevent dropout.
► This study will significantly contribute to the scientific evidence on the benefits of exercise and dietary support during chemotherapy in an understudied group of patients.
► This study offers a combined exercise and dietary intervention, and therefore the effects of an exercise or dietary intervention alone cannot be disentangled.

ovarian cancer-specific comorbidities and adverse effects of ovarian cancer and its treatment following the i3-S strategy.

**Ethics and dissemination** This study has been approved by the medical ethical committee of the Amsterdam UMC (reference: 018). Results of the study will be published in international peer-reviewed journals.

**Trial registration number** Netherlands Trial Registry (NTR6300).

## INTRODUCTION

Ovarian cancer is the seventh most common type of cancer worldwide, with 239 000 new cases annually.[1] Ovarian cancer is often

diagnosed at an advanced International Federation of Gynaecology and Obstetrics (FIGO) stage, resulting in a poor overall prognosis.[2] The overall 5-year survival rate for ovarian cancer is 30%–40%, but ranges from 92% in patients with FIGO stage I at diagnosis to 29% in patients with FIGO stage IV.[2]

The majority (90%) of malignant ovarian tumours are of epithelial origin.[2] Standard care for epithelial ovarian cancer includes cytoreductive surgery and platinum-based and taxane-based (neo)adjuvant chemotherapy.[3 4] As a consequence of ovarian cancer and its treatment, many women have physical and/or psychosocial problems such as reduced physical function and fatigue, compromising quality of life.[5–14] In addition, previous studies reported that half of the patients suffer from sarcopenia (ie, loss of skeletal muscle mass) or malnutrition at diagnosis and that the prevalence increased during neoadjuvant chemotherapy.[9 10 15 16] Independent from the presence of sarcopenia and malnutrition, studies reported that 24%–57% of patients with ovarian cancer are overweight and 10%–35% are obese.[9 10 17–21]

Observational studies in women with ovarian cancer found that sarcopenia, overweight and obesity at diagnosis; loss of body weight and muscle mass during treatment; and underweight after treatment are associated with a lower survival rate.[9–11 17–22] Furthermore, observational studies among patients with cancer, not including ovarian cancer, have shown that higher levels of physical activity or physical fitness are associated with better survival.[23 24] Therefore, it may be important to prevent weight gain or involuntary weight loss, and maintain physical fitness and muscle mass during treatment.

Both exercise and dietary interventions are non-pharmacological interventions that can positively influence body composition, physical fitness and function and reduce fatigue in patients with cancer.[25–29] A meta-analysis of intervention studies among the general population,[30] as well as the International Society of Sports Nutrition,[31] highlights that a combined exercise and dietary intervention is more effective for changing body composition than an exercise or dietary intervention alone. Most previous studies examining exercise or dietary interventions in patients with cancer were conducted in patients with breast cancer. It is unclear whether the effects of exercise and dietary interventions found in patients with breast cancer can be generalised to women with ovarian cancer. Compared with breast cancer, ovarian cancer is often detected in a more advanced stage[32] and in older women. Ovarian cancer also has a substantially different treatment trajectory, that is, different types of chemotherapy and other adjuvant therapy regimens. A few previous pilot studies have indicated that low-intensity to moderate-intensity exercise interventions[33 34] or a combined exercise and dietary intervention[35] during chemotherapy is feasible in women with ovarian cancer. Adequately powered randomised controlled trials (RCT) evaluating the effect of an exercise and dietary intervention during treatment in patients with ovarian cancer

are lacking. Therefore, the Physical Activity and Dietary intervention in OVArian cancer (PADOVA) study was initiated. The PADOVA study aims to evaluate the effectiveness of a combined exercise and dietary intervention during chemotherapy on body composition, physical function and fatigue as primary outcomes, compared with a usual care control group in women undergoing chemotherapy for ovarian cancer. Secondary outcomes are physical activity and fitness, dietary intake, body mass index (BMI), patient-reported outcomes, treatment toxicity and completion rates. The secondary aim is to conduct an extensive process evaluation to examine how and why the intervention is (in)effective.

To optimise intervention feasibility and study retention rates, we aim to offer exercise and dietary interventions that are specifically tailored to comorbidities and disease-induced and treatment-induced adverse effects that individual patients with ovarian cancer may face. This article presents the design of the multicentre randomised controlled PADOVA trial and describes how the combined exercise and dietary intervention can be tailored specifically to patients with ovarian cancer.

## METHODS AND ANALYSIS
The PADOVA study is a multicentre, single-blind RCT. Patient inclusion and data collection had started in February 2018 and are currently ongoing.

### Participants
The study aims to include 122 adult (aged ≥18 years) women who are scheduled for (neo)adjuvant first-line chemotherapy treatment for primary epithelial ovarian cancer. Patients are excluded from this study if they have had a prior cancer diagnosis within 5 years, are not able to perform basic activities of daily living, have a contraindication for exercise (eg, heart failure), have a cognitive disorder or severe emotional instability (eg, schizophrenia, Alzheimer), are unable to read and/or write Dutch, or have a life expectancy of less than 3 months.

### Recruitment and randomisation
Patients are recruited from the Center of Gynaecologic Oncology Amsterdam (which is a collaboration of all gynaecological oncologists from Amsterdam UMC, Netherlands Cancer Institute/Antoni van Leeuwenhoek and affiliated peripheral hospitals) and Catharina hospital and its collaborating peripheral hospitals in the south of the Netherlands. After diagnosis and before the start of neoadjuvant chemotherapy (±2 weeks) or adjuvant chemotherapy (±4 weeks), the gynaecological oncologist informs patients about the PADOVA study during an outpatient clinic visit. Written informed consent is obtained from all patients prior to participation. After baseline measurement, participants are stratified by FIGO stage (low (I/II) vs high (III/IV stage) and chemotherapy regimen (primary surgery followed by chemotherapy vs

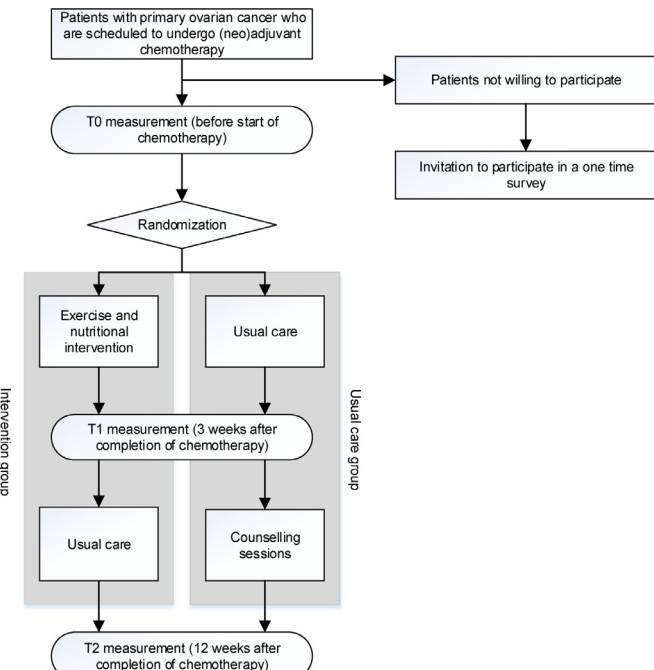

**Figure 1** Overview of the physical activity and dietary intervention in OVArian cancer study design and procedures.

neoadjuvant chemotherapy followed by interval debulking and adjuvant chemotherapy) and randomly allocated to either a combined exercise and dietary intervention or a usual care group. An independent researcher performs the randomisation using a table of random numbers in blocks of four generated by an independent statistician. Allocation sequence is concealed from the research and clinical staff. After randomisation, patients in both the intervention and control groups receive a brochure with general information on physical activity, diet and body weight recommendations for cancer survivors.[36] These recommendations are not individualised or supervised. Patients who do not wish to participate in the study are invited to complete a single questionnaire examining relevant characteristics and reasons for declining participation to verify representativeness of the study population. See figure 1 for an overview of the study design and procedures.

### Development and description of the combined exercise and dietary intervention

The aims of the exercise and dietary intervention are to maintain physical fitness and function, prevent the loss of lean body mass and maintain a healthy body weight during (neo)adjuvant chemotherapy. The intervention starts at the first cycle of chemotherapy and continues until 3 weeks after the last cycle. In general, patients receive six cycles of chemotherapy (duration of ±18 weeks). In case of dose delay or discontinuation of chemotherapy treatment, duration of the intervention and range of time between study measurements might differ between patients. For optimal intervention feasibility and study retention rates, it is important to offer an exercise

and dietary intervention in the PADOVA study, which is specifically tailored to patients with ovarian cancer. The exercise intervention was based on the exercise intervention that has previously been shown to effectively maintain physical fitness, limit fatigue and enhance quality of life during chemotherapy in patients with breast cancer.[28] The dietary intervention was based on Dutch and international guidelines on general nutritional support for patients with cancer and nutritional support for malnourished patients or patients with ovarian cancer.[37–41] These interventions were tailored via the i3-S strategy to ovarian cancer-specific comorbidities and adverse effects of ovarian cancer (eg, ascites) and its surgical and chemotherapy treatment. The i3-S strategy was introduced in 2015 by Dekker *et al*[42] to develop comorbidity-related adaptations to exercise therapy and has previously been used to tailor exercise interventions to potential comorbidities and adverse effects of breast cancer treatment[43] and to patients with knee osteoarthritis.[44]

The i3-S strategy consists of four steps, via which relevant information on the specific disease is collected. In the first step, information on comorbidities that occur in patients with ovarian cancer was gathered. All registered comorbidities of patients (n=109) who were treated for ovarian cancer in 2016 in the Amsterdam UMC were collected from patients records. Comorbidities were categorised according to International Classification of Diseases, 10th revision[45] (see table 1 for an overview of all comorbidities).

Comorbidities with a prevalence of ≥5% were included in steps 2–4 of the i3-S strategy. In addition to comorbidities, the potential adverse effects of ovarian cancer and its treatment were added to the inventory. These adverse effects were derived from the literature,[46] guidelines[47 48] and expert meetings with (gynaecological) oncologists. We included the adverse effects (incidence ≥1%) of carboplatin and paclitaxel as these are currently standard chemotherapy treatments for ovarian cancer.[4] In addition, we included all potential adverse effects of surgery and ovarian cancer itself. We focused on potential adverse effects relevant for healthcare providers delivering the exercise or dietary interventions. As a consequence, we excluded descriptions of acute adverse effects of chemotherapy monitored by treating physicians during admission to the hospital. In addition, we checked the overlap of comorbidities and adverse effects of ovarian cancer and its treatment with previously published i3-S papers for knee osteoarthritis[44] and breast cancer[43] or nutritional guidelines.[37–39 41] The following comorbidities (ie, hypertensive diseases, ischaemic heart diseases, other forms of heart disease, chronic lower respiratory diseases and diabetes mellitus) and adverse effects (clinical parameters such as leucopenia/neutropenia, trombopenia and anaemia, and symptoms of dyspnoea, nausea, vomiting or diarrhoea, skin and nail changes, fever, dizziness, decreased or increased heart rate, change in body weight, depression, numbness/loss of sensation, hearing and/or visual impairments, fatigue, pain and chest pain) were

| Comorbidity | % |
|---|---|
| **Table 1** Comorbidities in patients with ovarian cancer (n=109) | |
| Hypertensive diseases (ie, hypertension) | 28 |
| Ischaemic heart diseases (ie, angina pectoris, myocardial infarction (>6 months), percutaneous transluminal coronary angioplasty, coronary artery bypass grafting) | 6 |
| Other forms of heart disease (ie, atrial fibrillation and flutter, other cardiac arrhythmias, heart failure, other forms of heart disease not specified) | 9 |
| Cerebrovascular diseases (ie, stroke, not specified as haemorrhage or infarction) | 4 |
| Diseases of arteries, arterioles and capillaries (ie, aortic aneurysm, peripheral vascular diseases) | 3 |
| Diseases of veins, lymphatic vessels and lymph nodes, not elsewhere classified (ie, phlebitis and thrombophlebitis) | 1 |
| Chronic lower respiratory diseases (ie, chronic obstructive pulmonary disease, asthma) | 16 |
| Diabetes mellitus (unspecified) | 8 |
| Disorders of thyroid gland (ie, hypothyroidism/hyperthyroidism) | 6 |
| Disorders of other endocrine glands (ie, hypoparathyroidism/hyperparathyroidism) | 1 |
| Episodic and paroxysmal disorders (ie, transient ischaemic attack) | 1 |
| Other disorders of the nervous system (unspecified) | 1 |
| Dementia in other diseases classified elsewhere (ie, dementia in Parkinson's disease) | 1 |
| Diseases of liver (unspecified) | 1 |
| Disorders of gallbladder, biliary tract and pancreas (ie, cholelithiasis) | 5 |
| Renal failure (unspecified) | 1 |
| Soft tissue disorders (ie, rheumatism, unspecified) | 2 |
| Tuberculosis | 1 |
| Malignant neoplasms (excluding basal cell carcinoma) | 7 |

described in previous publications[43 44] or nutritional guidelines[37–39 41] and are not presented in this article.

In the second and third step, contraindications and restrictions on exercise training were gathered, as well as solutions for exercise training in ovarian cancer-specific comorbidities and adverse effects of ovarian cancer and its treatment. This was based on literature,[42–44 49] guidelines[38] and/or expert opinions of (gynaecological) oncologists and physical therapists specialised in oncology.

In the final step, comorbidities, adverse effects of ovarian cancer and its treatment, and also exercise contraindications and restrictions were translated into clinical parameters and symptoms that can be monitored during the intervention of the PADOVA study. All information was synthesised in a framework (table 2).

### Exercise intervention

The exercise intervention consists of two 1-hour exercise sessions per week, including moderate-intensity to high-intensity resistance and aerobic exercises. The exercise sessions are supervised by a physical therapist specifically trained in treating oncology patients, to inform patients on and monitor appropriate and safe exercise strategies. These physical therapists are affiliated with a nationwide network that includes >600 physical therapy practices. This enables to offer the intervention close to a patient's home. Depending on physical therapy practice, patients train in small groups with other (non-)PADOVA patients (with cancer).

The exercise intervention is based on the exercise training principles (ie, specificity, progression, overload, initial values, reversibility and diminishing returns).[50] The sessions start with a warming up of 10 min. Resistance exercises targeting six large muscle groups are conducted for 20 min per session. Prescribed exercises include vertical row, leg press, bench press, pullover, abdominal crunch and lunge. Due to the abdominal wound in the postoperative period (4–6 weeks), eccentric exercises with abdominal muscles and pressure on the abdomen are prevented by omitting heavy lifting and exercises such as abdominal crunch and pullover. Instead, exercises with an isometric use of abdominal muscles such as lateral raise or leg extension are performed. An overview of all adaptations for patients with ovarian cancer is shown in table 2. One repetition maximum (1RM) testing is repeated every 3 weeks, in line with chemotherapy regimen, to ensure adequate training intensity over time. The load of each resistance exercise is 70%–80% of the 1RM with a gradual increase per week in between 1RM testing. Exercises are performed in two sets of 8–10 repetitions. When the participant is unable to perform two sets of 10 repetitions or when the Borg Scale of perceived exertion exceeds 15, the load will be decreased by one step. When the Borg Scale of perceived exertion decreases to <12, the load will be increased. Aerobic exercises are conducted for 30 min per session, with an intensity of 50%–80% (gradually increasing) of the maximal workload as estimated by the steep ramp test.[51] This test is repeated every 6 weeks to ensure adequate workload over time. When the Borg Scale of perceived exertion decreases to a score of ≤12 or increases to ≥16, the workload is adjusted.[52] In addition to the supervised sessions, patients are encouraged to be physically active on at least 3 additional days a week for 30 min to meet the recommended physical activity levels.[40]

### Dietary intervention

The dietary intervention is provided by oncology dietitians once every 3 weeks during face-to-face sessions of 30–45 min each at the hospital or by telephone using motivational interviewing techniques.[53] Motivational interviewing is an effective counselling method for achieving health behaviour change using techniques such as reflective listening and summarising, focusing on what the patient wants, thinks and feels.[54] The PADOVA dietary intervention is based on Bandura's social cognitive theory (SCT).[55] Lifestyle interventions

**Table 2** Framework with alternations for the exercise and dietary intervention

| Comorbidities and adverse effects of ovarian cancer and its treatment | Considerations | Actions/strategy |
|---|---|---|
| **Comorbidities** | | |
| Disorders of the thyroid gland (ie, hypothyroidism/hyperthyroidism) | Consider the following complications of disorders of thyroid gland: | |
| | ► Weight loss or weight gain | ► Refer to a dietitian when weight loss/weight gain occurs |
| | ► Bradycardia in hypothyroidism or tachycardia in hyperthyroidism | ► Explain to patient bradycardia/tachycardia due to hypothyroidism/hyperthyroidism |
| | | ► Monitor symptoms: in case of persistent coexisting symptoms (dyspnoea, anxiety, fatigue), terminate exercise and refer to a physician |
| | ► Low energy/fatigue in hypothyroidism | ► Explain to patient fatigue due to hypothyroidism |
| | | ► Refer to a physician when fatigue does not reduce in a few weeks |
| Disorders of gallbladder, biliary tract and pancreas (ie, cholelithiasis) | | ► Monitor symptoms and in case of persistent pain, terminate exercise and refer to a physician |
| Malignant neoplasms (excluding basal cell carcinoma)* | Consider comorbidities and adverse effects of malignant neoplasm that might interfere with the intervention | |
| **Adverse effects of ovarian cancer** | | |
| Ascites (60%) | Consider the following complications of ascites: | |
| | ► Discomfort | ► Adjust exercise to a comfortable intensity or posture |
| | ► Pleural effusion and/or shortness in breath | ► In case of disproportional dyspnoea, terminate exercise session and refer to a physician |
| | ► Diminished nutritional intake | ► Refer to a dietitian |
| **Adverse effects of chemotherapy and surgery** | | |
| Abdominal wound | | ► In the postoperative period (4–6 weeks), exercise is allowed, but pressure on the abdomen should be avoided; therefore, the 'abdominal crunch' and 'pullover' could be replaced with a 'lateral raise' and 'leg extension' |
| | | ► After 4–6 weeks, isometric exercises could be replaced by eccentric exercises |
| | | ► Monitor symptoms: in case of pain or discomfort, decrease training intensity or resistance |
| | Consider the following complications of an abdominal wound: | |
| | ► Fever >38.5°C due to wound infection | ► Contraindication for exercise, refer to a physician |
| Intestinal stoma | | ► See recommendation under 'abdominal wound' |
| | | ► Avoid contact sport (eg, football or martial arts) |
| | Consider the following complications of an intestinal stoma | |
| | ► High-output stoma (production of >1 L per 24 hours during ≥3 days) | ► Contraindication for exercise, refer to a physician and/or dietitian |
| (Risk of) lymphoedema in legs | | ► Monitor leg volume during exercise programme; ask for symptoms of lymphoedema |
| | | ► Refer to a lymphoedema specialist when lymphoedema is present or when leg volumes increases and symptoms arise (advice on how to progress with exercise) |
| | Consider the following complications of lymphoedema in legs: | |
| | ► Numbness/loss of sensation | ► Be careful with exercises that include walking, running or balance to prevent falls |
| | | ► Advise patient to wear good-fitting, stable footwear with good grip under surface |

Continued

**Table 2** Continued

| Comorbidities and adverse effects of ovarian cancer and its treatment | Considerations | Actions/strategy |
|---|---|---|
| Deep vein thrombosis (in leg) | | ► Contraindication for exercise, refer to a physician when the following symptoms occur: pain in the leg, red or discoloured skin or a feeling of warmth |
| Thrombophlebitis | | ► Avoid pressure or impact on affected area |
| | | ► Monitor symptoms and in case of pain or discomfort decrease training intensity or resistance |
| Nervousness/confusion | Consider the following causes of nervousness/confusion | |
| | ► Severe anxiety or a psychiatric disorder | ► Give the patient time to discuss feelings or thoughts |
| | | ► Contraindication for exercise, refer to a physician when a serious psychiatric disorder might be present |
| Gastrointestinal symptoms (ie, anorexia, dyspepsia, constipation, taste disorder, dry mouth, mouth ulcers, stomatitis) | Consider the following complications of gastrointestinal symptoms: | |
| | ► Diminished nutritional intake | ► Refer to a dietitian and/or physician |
| Melaena | | ► Contraindication for exercise, refer to a a physician |

*Patients with a current other malignancy or prior diagnosis (within 5 years) of cancer were excluded from participation in the Physical Activity and Dietary intervention in OVArian cancer study.

based on SCT have been shown to improve health behaviours in patients during and after cancer treatment[56] and generally focus on improving self-efficacy, dealing with sociostructural factors (impediments/barriers and facilitators), managing outcome expectations and setting goals to improve health behaviours.[55] The behaviour change techniques (BCTs) used to promote health behaviour change are defined according to BCT Taxonomy V.1[57] and are listed in online supplemental table A. The content of the dietary intervention is based on the dietary guideline set by the World Cancer Research Fund (WCRF)/American Institute for Cancer Research (AICR)[40] and a sufficient total protein intake of at least 1.2 g of protein per kilogram body weight per day[41 58] and 25 g of protein per meal, since such evenly distributed protein intake is expected to optimise muscle protein synthesis.[59] Counselling is tailored to the nutritional needs of each individual patient according to body composition, nutritional status and dietary intake during chemotherapy. Patients who are at risk of malnutrition are primarily counselled for prevention of weight loss by maintaining sufficient caloric intake, particularly protein intake. Patients who are not at risk of developing malnutrition are primarily counselled to meet the dietary guidelines set by the WCRF/AICR. During each counselling session, patients will receive feedback on their body weight, BMI, body composition assessed via bioelectric impedance analysis (BIA), diet quality and the extent to which they meet the protein goals and WCRF/AICR dietary recommendations. Extended information on the content per dietary counselling session is provided in online supplemental table B.

### Control group
Women in the control group will receive usual care during chemotherapy, which includes referral to a dietitian when malnutrition is detected by the gynaecological oncologist. Usual care does not include structured exercise and/or dietary counselling. To prevent non-participation and dropout, patients in the control group are offered a maximum of three exercise and three dietary counselling sessions in 12 weeks after completion of chemotherapy and the first follow-up measurement.

### Outcome measurements
An investigator blinded from group allocation conducts measurements at three time points. Participants are instructed not to reveal their group allocation. Baseline measurements are conducted before randomisation and at the start of chemotherapy (T0), the second measurement at 3 weeks after completion of chemotherapy (T1) and the last measurement (T2) at 12 weeks after T1. An overview of all outcome measurements is presented in table 3.

### Primary outcomes
Primary outcomes are body composition, physical function and fatigue. The body composition is assessed via skeletal muscle area and fat mass. The skeletal muscle area is assessed at T0 and T1 using routine CT imaging (first image extending from the third lumbar vertebra to iliac crest) conducted for diagnostic purposes. CT is considered the gold standard for assessment of muscle mass in patients with cancer.[60] Because the CT of the third lumbar vertebra is not yet validated for the assessment of body fat mass,[61] fat mass is assessed with a non-invasive

**Table 3** Summary of outcome measurements

| Outcome | Instrument | T0 | T1 | T2 |
|---|---|---|---|---|
| **Primary outcomes** | | | | |
| Body composition | CT imaging for the assessment of skeletal muscle area | X | X | |
| | Bioelectric impedance analysis for the assessment of fat mass | X | X | X |
| Physical fatigue | Multidimensional Fatigue Inventory[64] | X | X | X |
| Physical function | European Organisation for Research and Treatment of Cancer Quality of Life Questionnaire Core 30[63] | X | X | X |
| **Secondary outcomes** | | | | |
| Physical activity | Accelerometer (actigraph) for objective measurement of physical activity in 7 consecutive days during all waking hours | X | X | X |
| | Physical Activity Scale for the Elderly[78] for self-reported levels of physical activity | X | X | X |
| Physical fitness | Maximum oxygen uptake (peak VO$_2$) during a maximum exercise test on a cycle ergometer using a ramp protocol for the assessment of cardiorespiratory fitness | X | X | X |
| | Handheld dynamometer for the assessment of muscle strength | X | X | X |
| Body mass index | Body height on a calibrated scale to the nearest 0.1 mm | X | X | X |
| | Body weight on a calibrated scale to the nearest 0.1 kg | X | X | X |
| Dietary intake; WCRF/AICR guidelines | Food frequency questionnaire (developed by Wageningen University) | X | X | X |
| Dietary intake; protein intake | Self-composed food frequency questionnaire | X | X | X |
| Health-related quality of life and symptoms | European Organisation for Research and Treatment of Cancer Quality of Life Questionnaire–Ovarian Cancer Module 28[79] | X | X | X |
| | European Organisation for Research and Treatment of Cancer Quality of Life questionnaire–Chemotherapy-Induced Peripheral Neuropathy 20-Item Scale[80] | X | X | X |
| | Hospital Anxiety and Depression Scale[81] | X | X | X |
| | Pittsburgh Sleep Quality Index[82] | X | X | X |

Continued

**Table 3** Continued

| Outcome | Instrument | T0 | T1 | T2 |
|---|---|---|---|---|
| Chemotherapy therapy completion rates and treatment toxicity | Medical records of which chemotherapy completion rates will be assessed as the relative dose intensity, that is, the amount of particular chemotherapy given in relation to the originally planned chemotherapy dose | | X | |
| **Other study parameters** | | | | |
| Blood sampling | Venous blood sample (40 mL) | X | X | |
| Smoking and sociodemographics | Self-composed questionnaire | X | | |
| Contamination of control group* | Self-composed questionnaire | | X | |

*Parameter will only be assessed in the control group.
AICR, American Institute for Cancer Research; WCRF, World Cancer Research Fund.

measurement BIA.[62] Physical function is assessed using the physical function subscale of the validated European Organisation for Research and Treatment of Cancer Quality of Life Questionnaire Core 30.[63] Physical fatigue is assessed with the physical fatigue subscale of the Multidimensional Fatigue Inventory.[64]

### Secondary outcomes
Secondary outcomes are physical activity, physical fitness (ie, cardiorespiratory fitness and muscle strength), dietary intake, BMI, health-related quality of life and symptoms of neuropathy, anxiety and depression, sleep disturbances, chemotherapy treatment toxicity and completion rates. The measurement instruments for the assessment of the secondary outcomes are presented in table 3.

### Other study parameters
During the T0 and T1 visit, a venous blood sample is drawn and stored in the PADOVA biobank for future biomarker studies (eg, to assess immune system functioning). Because collection of blood samples is an addition to participation in the PADOVA study, an additional informed consent is obtained. Covariates such as clinical data (eg, cancer subtype, FIGO stage) and sociodemographic characteristics (eg, age, ethnicity) are assessed at baseline. Contamination (ie, received supervised exercise and/or dietary counselling) of the control group is assessed by a questionnaire at T1.

### Process evaluation
A process evaluation is conducted using a mixed-methods approach (using both quantitative and qualitative research methods). Quantitative data on behavioural counselling skills of dietitians are assessed during counselling sessions with the Behaviour Change Counselling Inventory.[65] Patients in the intervention group fill out the Health Care Climate Questionnaire to assess the extent to

which they perceive an autonomy-supportive environment during counselling.[66] Quantitative data on behavioural determinants of physical activity and dietary behaviour (ie, outcome expectations, self-efficacy for eating and exercise habits, sociostructural factors, stage of change, type of motivation and knowledge) are assessed on T0, T1 and T2 by questionnaires in all patients. The following process evaluation components are examined by physical therapists and dietitians and reported in a form: dose delivered, dose received and fidelity. Acceptability of the intervention is assessed by a questionnaire filled out by patients and semistructured interviews among patients, physical therapists and dietitians conducted by a researcher. Interviews will be transcribed verbatim, coded in several phases[67 68] and analysed with Atlas.ti 8.4.20. Analyses are partly performed concurrently with data collection because interviews will be held until data saturation is reached. An overview of all measurement instruments for process evaluation is presented in table 4.

### Sample size calculation

Sample size calculation is based on the results of a previous RCT among patients with breast cancer that evaluated the effects of a combined aerobic and resistance exercise intervention (similar to the PADOVA study),[28] a pilot study among patients with ovarian cancer that evaluated the feasibility of an exercise intervention[34] and clinically relevant differences in body composition and peak oxygen uptake.[34 69–74] With 53 patients per study-arm, we are able to detect a clinically relevant between-group difference in effects directly after intervention on physical function (10 points), physical fatigue (2.7 points), body composition (3% in percentage body fat and muscle mass) and 10%–15% difference in peak oxygen uptake (alpha=0.05; power=0.80). Taking into account a dropout of 15%, we aim to include 61 patients per group. Dropout rates are based on the 10%–15% dropout rates from previously conducted Dutch exercise trials in patients with cancer.[27 28 75]

### Statistical analysis

The primary analysis focuses on the effects of the intervention directly after completion of chemotherapy (T1), since both the intervention and control groups have received counselling from a physical therapist and/or dietitian at follow-up (T2). Data will be analysed according to the intention-to-treat principle.

Intervention effects (at T1) will be assessed using linear regression analysis for continuous outcomes by regressing the intervention on post-test value (T1) of the outcome, adjusting for baseline values (T0). To examine whether intervention effects on body composition, physical function or fatigue are mediated by changes in physical activity and fitness, and/or diet, a series of regression analysis according to the product-of-coefficients test will be conducted.[76] Potential effect modification by relevant demographic (eg, age) and clinical (eg, cancer stage, treatment regimen) variables will be explored by adding

the variable and its interaction term with the intervention into the regression model.

### DISCUSSION

This article presents the rationale and design of the PADOVA study and the procedure of tailoring an exercise

**Table 4** Overview of all measurement instruments for process evaluation

| Outcome | Instrument | T0 | T1 | T2 |
|---|---|---|---|---|
| Process evaluation | | | | |
| Behavioural determinants | Multidimensional Outcome Expectations for Exercise Scale[78] | X | X | X |
| | Self-efficacy for eating and exercise habits—self-composed items | X | X | X |
| | Sociostructural factors, self-composed items | X | X | X |
| | Stage of change[83] | X | X | X |
| | Type of motivation: Treatment Self-Regulation Questionnaire[84] | X | X | X |
| | Knowledge (on WCRF recommendations): Items used in previous study[85] | X | X | X |
| Acceptability of intervention | Self-composed items* | | X | |
| | The Healthcare Climate Questionnaire–Short Form (six items)[66] | | X | |
| Acceptability of intervention | Semistructured interviews in healthcare professionals† | | X* | X‡ |
| | Semistructured interviews in participants | | X* | X‡ |
| Dose delivered | Checklist†—Amount of (components of) sessions provided by physical therapist/dietitian | Daily, during intervention | | |
| Dose received | Checklist†—Amount of (components of) sessions participants actively engaged in | Daily, during intervention | | |
| Fidelity | Checklist†—Extent to which the intervention was executed as prescribed in the protocol (eg, reasons for and amount of adaptations to the protocol) | Daily, during intervention | | |
| Behavioural counselling skills§ | Behaviour Change Counselling Inventory[65] | During intervention | | |

*Parameters will only be assessed in the intervention group.
†Parameter will be examined in physical therapists and dietitians.
‡Parameter will only be assessed in the control group.
§Parameter will be examined in dietitians.
WCRF, World Cancer Research Fund.

and dietary intervention to patients with ovarian cancer. The PADOVA study aims to examine the effectiveness of a combined exercise and dietary intervention on body composition, physical function and fatigue in patients with ovarian cancer during chemotherapy. In addition, a process evaluation is conducted to examine the effective and ineffective components of the PADOVA intervention. A strength of this study is its randomised controlled design in women with ovarian cancer and systematic development of the exercise and dietary intervention by the i3-S strategy to ensure adequate tailoring of the intervention to this patient-specific group. Because the intervention is adjusted to comorbidities and disease-induced and treatment-induced effects, we expect compliance to the intervention to be high and dropout to be low. Another strength is process evaluation because it retrieves information on how and why the intervention is (un)successful.[77] With this information, the exercise and dietary intervention will be improved before the combined intervention is implemented in clinical practice.

Aiming to maximise benefits on body composition, we combined an exercise and dietary intervention. Consequently, we are unable to disentangle the effects of the exercise and dietary intervention. However, we plan to perform a mediation analysis to explore whether the intervention effects on body composition can be explained by changes in exercise or dietary components. Another limitation of this study is its inability to study long-term effects of the intervention compared with the control group. In the PADOVA study, exercise and dietary counselling will be offered to the control group after the first follow-up measurement to prevent non-participation and dropout and to limit contamination (exercise and dietary intervention) in the control group during chemotherapy.

Close collaboration with physical therapists specifically educated to supervise patients with cancer, affiliated with a network throughout the Netherlands, has the advantage that the exercise intervention can be offered close to a patient's home. This increases the reach and scalability of the intervention. Consequentially, this might also lead to small variations in the implementation of the intervention protocol by each individual physical therapist. This study will contribute to the evidence on the potential benefits of an exercise and dietary intervention in patients with ovarian cancer during chemotherapy treatment who often face a complex and unfavourable disease trajectory. If proven effective, a combined exercise and dietary intervention for patients with ovarian cancer can be implemented in clinical practice.

## Patient and public involvement

Patients were involved in the development of study-specific patient information, and they were asked to assess the burden of time required to participate in this study. The opinion of patients has been considered to improve the readability of the patient information sheets on the study. During the study, patients will be interviewed (eg, on acceptability of the intervention) as part of the process evaluation. This information could be important for implementation of the intervention in clinical practice. Study results will be presented to patients in collaboration with the patient community.

## ETHICS AND DISSEMINATION

This study was approved by the medical ethical committee of the Amsterdam UMC (reference: 018). Additional approval was obtained for the participating hospitals. The trial is registered in the Netherlands Trial Register. Signed informed consent is required from all included participants. Results of the study will be published in international peer-reviewed journals.

**Author affiliations**
[1]Epidemiology and Biostatistics, Amsterdam Public Health research institute, Cancer Center Amsterdam, Amsterdam UMC, Vrije Universiteit Amsterdam, Amsterdam, The Netherlands
[2]Physiology, Radboud Institute of Health Sciences, Radboudumc, Nijmegen, The Netherlands
[3]Medical and Clinical Psychology, Center of Research on Psychology in Somatic diseases, Tilburg University, Tilburg, The Netherlands
[4]Obstetrics and Gynaecology, Cancer Center Amsterdam, Center for Gynaecologic Oncology Amsterdam (CGOA), Amsterdam UMC, Vrije Universiteit Amsterdam, Amsterdam, The Netherlands
[5]Obstetrics and Gynaecology, Amsterdam UMC, Univ(ersity) of Amsterdam, Amsterdam, Netherlands
[6]Gynaecology, Center for Gynaecologic Oncology Amsterdam (CGOA), The Netherlands Cancer Institute - Antoni van Leeuwenhoek Hospital, Amsterdam, Netherlands
[7]Human Nutrition and Health, Wageningen University and Research, Wageningen, The Netherlands
[8]Rehabilitation Medicine, Amsterdam UMC, Vrije Universiteit van Amsterdam, Amsterdam, The Netherlands

**Contributors** EK, GGK, LMB and MH conceived the study. GGK, LMB, LRCWvL, MH and SS designed the study. LMB, MH, RJH and SS designed the intervention. LMB, MH and SS wrote the manuscript. All authors read and approved the final manuscript.

**Funding** The PADOVA study is funded by the Dutch Cancer Society, grant number VU 2015-7950. The Dutch Cancer Society was not involved in the design of the study, collection, analysis and interpretation of data, nor in writing the manuscript.

**Competing interests** None declared.

**Patient consent for publication** Not required.

**Provenance and peer review** Not commissioned; externally peer reviewed.

**ORCID iD**
LM Buffart http://orcid.org/0000-0002-8095-436X

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
