## [Reviewer comments · BMJ Open]

ARTICLE DETAILS

TITLE (PROVISIONAL)	Rationale and study protocol of the Physical Activity and Dietary intervention in women with OVArrian cancer (PADOVA) study: A randomised controlled trial to evaluate effectiveness of a tailored exercise and dietary intervention on body composition, physical function and fatigue in women with ovarian cancer undergoing chemotherapy
AUTHORS	Stelten, Stephanie; Hoedjes, Meeke; Kenter, Gemma; Kampman, Ellen; Huijsmans, Rosalie; van Lonkhuijzen, Luc; Buffart, LM

VERSION 1 - REVIEW

REVIEWER	Brian C Focht The Ohio State University USA
REVIEW RETURNED	06-Mar-2020

GENERAL COMMENTS	This protocol paper addresses the rationale and design of the PADOVA RCT examining the efficacy of a tailored exercise and dietary intervention for ovarian cancer (OVCA) patients undergoing chemotherapy. The trial is generally well-designed and addresses a vulnerable, understudied sample of OVCA patients on chemotherapy who are at risk for unfavorable shifts in body composition and functional decline. Given OVCA patients could benefit from lifestyle intervention combining exercise and dietary components, the proposed trial is significant, reasonable novel, and has potential for meaningful impact. Despite these notable strengths of the study rationale and design, there are multiple conceptual and methodological concerns which detract from the potential impact of the paper. While most concerns reflect issues which simply require clarification, other select concerns necessitate more detailed justification or reinterpretation. 1. As it is well-established that theory-based interventions are superior to atheoretical and/or theory-informed approaches, the theoretical foundation for the lifestyle intervention should be explicitly presented. Additionally, how theory informed the intervention design and selection of key process and outcome measures should also be provided.2. A more detailed description of the recruitment procedures should be provided. Explicitly addressing the process of how the clinical referrals are obtained would be instructive.3. A more detailed description of the progressions implemented in
---

	the exercise prescription is also warranted. Specifically, the description of how load, volume, and volume-load progressions of the resistance exercise component are implemented is underdeveloped and further detail of this procedure would be particularly informative. 4. Description of the specific strategies to promote adoption and maintenance of change in dietary intake are not sufficiently developed and could be more clearly articulated. 5. The 15% drop out rate and anticipated effect size for body composition ($d = .55$) both appear overly optimistic and should be more explicitly justified. From a conceptual perspective, the intervention appears to involve considerable contact with specialists with expertise in exercise physiology, physical therapy, and dietetics. I urge the authors to explicitly address the implications of this approach for the reach, scalability and overall impact of the proposed intervention.
--	---

REVIEWER	Alexander R. Lucas Virginia Commonwealth University School of Medicine Department of Health Behavior and Policy, Department of Internal Medicine - Division of Cardiology Pauley Heart Center Virginia, United States
REVIEW RETURNED	09-Mar-2020

GENERAL COMMENTS	General comments The current study reports on the study protocol for a randomized controlled trial of physical activity and dietary intervention for women with ovarian cancer undergoing chemotherapy. The study will test and compare the effects of a tailored exercise and dietary intervention on primary outcomes of body composition, physical function, and fatigue. The study will also examine and compare the effects of these interventions on a battery of secondary outcomes assessed before starting chemotherapy, 3 weeks after completing chemotherapy and 12 weeks later. The aims of the manuscript include describing the tailoring of PA and Diet to the comorbidity status and presence of adverse events (AE's) in patients. The proposed study enrolled its first patient in February 2018 and is currently open for patient inclusion targeting a total of 122 patients by December 2020. The target population is epithelial ovarian cancer patients scheduled to receive (neo) adjuvant chemotherapy. The interventions are designed to be 2 x 1-hour supervised sessions per week, including both resistance and aerobic exercise. Dietary counseling includes individually tailored 30-minute sessions every 3 weeks during chemotherapy. Waitlisted control participants will receive education and information about diet and activity with access to counseling sessions upon request following the first assessment. The study is relatively novel in that it will examine the effectiveness of a tailored combination lifestyle intervention (shown to be effective in some studies) in an understudied cancer population undergoing
--

chemotherapy. A secondary aim of the study is to evaluate the intervention processes. I commend the authors for preparing this study and for the detail included in the reporting of the protocol. Strengths of the study are the randomized design, the use of extensive objective measures of outcomes, detailed process on tailoring the intervention (i3-s) and process evaluation. A limitation of the design is the lack of long term follow up due to the control group being provided with access to a limited number of intervention sessions.

Thank you for the opportunity to review your work and I wish you luck with the study. I have only a limited number of queries.

Specific comments

Abstract

1. None

Introduction

2. None

Methods and analysis

3. Under the description of i3-S methodology, on page 9, lines 187-196, the authors state that previous papers have discussed comorbidities and AE's and that they will therefore not be included in this paper. Just to clarify—you will still refer to these guidelines (previously published) when determining tailoring considerations for your ovarian population, you are just not writing about them in this paper?

4. Under the Exercise Intervention description, lines 226-228, the authors describe the recommendation to include physical activity independently at least 3 additional days per week. Given the listed (table 1) comorbidities in ovarian cancer patients, how do the researchers plan to monitor the safety of patients in the home or unsupervised-setting? Specifically as regards to potential cardiovascular complications.

5. Can the authors provide, from their pilot study, information about the typical time between diagnosis (histological confirmation of ovarian cancer) and decision to treat with chemotherapy? While this may be reported in the feasibility paper you cite, and vary across settings/countries, it is important to consider the feasibility of enrolling patients to study during this demanding window. Any information about expected timeframes would be beneficial to readers and researchers.

Discussion

6. None.

Tables and Figures

7. None.

REVIEWER	Liz Steed Queen Mary University of London UK
REVIEW RETURNED	10-Mar-2020

GENERAL COMMENTS	This study describes an evaluation of a physical activity and dietary intervention for women with ovarian cancer. This is an important topic as although the benefits of such interventions have been demonstrated in other areas of cancer, primarily breast cancer, there has been little work in ovarian cancer. The paper is mainly well written, although a proof read to ensure correct grammar and English usage would be advised – in particular use of tense. As this is written as a protocol paper the future tense will most commonly be needed. There are a number of recommendations which I would encourage the authors to consider:- Background  - I find the presentation of the dietary data somewhat confusing and would encourage the authors to review the presentation of this information. For example it is suggested that a high percentage of individuals are overweight or obese, but yet malnutrition is also common. Is there any overlap between these groups or is there an additional underweight group. Weight lost is suggested to be associated with lower survival rate but is this true regardless of baseline BMI or are there benefits in say the overweight group from losing weight but risks for an originally underweight group. Exercise is suggested to be helpful in preventing weight gain but the above is not suggestive that this is a common problem with ovarian cancer. Methods  - My key concern with the study is the potential for contamination or lack of difference in intervention received between the groups. Both groups receive a brochure on diet, exercise and weight recommendations and the control group get an intervention after T1, the significant limitation of this should be made clearer and suggests more of a feasibility design than large scale RCT as implied in the introduction - More detail should be provided on the development of the intervention and description of the intervention, including possible theory and logic model if relevant (see below). Was PPI included in intervention development? - Whilst it is important that co-morbidities were considered and accounted for in the exercise intervention I do not feel the level of detail that is provided here, including 2 detailed tables is of particular interest. I would have thought a couple of lines explaining the strategy and a brief table showing common adjustments would be sufficient in the main text with more detailed discussion including tables 1 and 2 in supplementary information if the authors feel this is really necessary. - In contrast there appears to be very little detail on the exercise and dietary interventions including how these were delivered and how
---

they were developed.

For example it is important to know for the exercise intervention what the training of the physical therapist was, was exercised delivered in groups, if so what size were groups were these closed or open, were there any behavioural elements to promote exercise. Also how long was exercise delivered for – throughout chemotherapy? In which case was this a standard period or was it variable

- For the dietary intervention what motivational interviewing techniques were used, what was the training of the dieticians in these therapies

-

Control group – it is noted that individuals did not receive structured exercise or dietary counseling, however if they were malnourished at baseline would this not be addressed in standard care?

- it is not clear what the range of times T1 could be collected – are there different periods chemotherapy is given for. Is T2, 12 weeks after t1, or baseline?

- I am unclear what the purpose of the blood sample for a biobank is. Was consent for this a requirement for consent to the main study. A statement on ethical approval for this as well as other aspects of the study is important.

Process Evaluation

- It is encouraging to see a process evaluation included, however this is not completely congruent with the intervention description which provides little consideration of psychosocial factors. If, and if so how, these factors were targeted in the intervention should be made clear, including a logic model and description of underlying theory. If these factors were not targeted by the intervention a rationale for why these are being evaluated as mediator variables would be helpful. In contrast if the variables are not targeted by the intervention but are being considered as predictor variables this should be made clear and moderator analysis considered rather than mediator analysis.

- There is a statement about fidelity being measured, again this is encouraging however fidelity of what should be stated as should how it will be measured and analysed.

- it is stated that interviews will be analyzed by a data analysis program. Which program will this be? Do the authors mean a program will be used to assist the data analysis process rather than program actually do the analysis?

-

Discussion

- I feel it is overselling the study to call it a large RCT, the control group will be contaminated post intervention so no medium or long term outcomes can be examined. I appreciate this is recognized as a limitation however this presentation should be more balanced.

- Also although the study calls itself multicentre only one centre is

	named for recruitment and it appears a single physical therapist is used to deliver the intervention. Being more reflective on this and the implications of what would be needed for full implementation (including training programmes) if successful would be helpful.
--	--

VERSION 1 – AUTHOR RESPONSE

Reviewer: 1

This protocol paper addresses the rationale and design of the PADOVA RCT examining the efficacy of a tailored exercise and dietary intervention for ovarian cancer (OVCA) patients undergoing chemotherapy. The trial is generally well-designed and addresses a vulnerable, understudied sample of OVCA patients on chemotherapy who are at risk for unfavorable shifts in body composition and functional decline. Given OVCA patients could benefit from lifestyle intervention combining exercise and dietary components, the proposed trial is significant, reasonable novel, and has potential for meaningful impact. Despite these notable strengths of the study rationale and design, there are multiple conceptual and methodological concerns which detract from the potential impact of the paper. While most concerns reflect issues which simply require clarification, other select concerns necessitate more detailed justification or reinterpretation.

Author response:

We thank the reviewer for the compliments on the significance and design of the PADOVA trial.

1. As it is well-established that theory-based interventions are superior to a theoretical and/or theory-informed approaches, the theoretical foundation for the lifestyle intervention should be explicitly presented. Additionally, how theory informed the intervention design and selection of key process and outcome measures should also be provided.

Author response:

We agree with the reviewer that it is important to include information on the theoretical background of our intervention informing intervention content and (process) outcome measures.

The aim of the exercise and dietary intervention is to maintain physical fitness and function, to prevent the loss of lean body mass and to maintain a healthy body weight during (neo)adjuvant chemotherapy. Due to its focus on maintaining physical fitness during treatment, the exercise intervention is primarily based on the exercise training principles (i.e. specificity, progression, overload, initial values, reversibility and diminishing returns).[1] The dietary intervention uses Motivational Interviewing techniques [2] and is based on Bandura's Social Cognitive theory.[3]

We have now incorporated this information and a more detailed description of the PADOVA intervention in the manuscript (printed in bold).

Lines 218-219 on the theoretical foundation of the exercise intervention: 'The exercise intervention is based on the exercise training principles (i.e. specificity, progression, overload, initial values, reversibility and diminishing returns).[1]

Lines 220-232 on resistance training: 'Resistance exercises targeting six large muscle groups are conducted for 20 minutes per session. Prescribed exercises include vertical row, leg press, bench press, pull over, abdominal crunch and lunge. Due to the abdominal wound in the post-operative period (4-6 weeks) eccentric exercises with the abdominal muscles and pressure on the abdomen were prevented by omitting heavy lifting and exercises such as the abdominal crunch and pullover. Instead, exercises with an isometric use of the abdominal muscles such as a lateral raise or leg extension, are performed. One repetition maximum (1RM) testing is repeated every three weeks, in line with chemotherapy regimen, to ensure adequate training intensity over time. Load of each resistance exercise is 70-80% of the 1RM with a gradual increase per week in between one repetition maximum testing. Exercises are performed in 2 sets of 8-10 repetitions. When the participant is unable to perform 2 sets of 10 repetitions, or when the Borg Scale of perceived exertion exceeds 15, training load will be decreased by one step. When the Borg Scale of perceived exertion decreases to <12, the load will be increased.

Lines 232 – 235 on aerobic exercise: 'Aerobic exercises are conducted for 30 minutes per session, with an intensity of 50-80% (gradually increasing) of the maximal work load as estimated by the steep ramp test.[4] This test is repeated every 6 weeks to ensure adequate work load over time. When the Borg Scale of perceived exertion decreases to a score of ≤ 12 or increases to a score of ≥ 16 the work load is also adjusted'.

Lines 241-249 and lines 260-261 for dietary intervention:

"Motivational interviewing is an effective counselling method for achieving health behavior change using techniques as reflective listening and summarizing, focusing on what the patient wants, thinks and feels.[5] The PADOVA dietary intervention is based on Bandura's Social Cognitive Theory (SCT).[3] Lifestyle interventions based on SCT have been shown to improve health behaviours in patients during and after cancer treatment,[6] and generally focus on improving self-efficacy, dealing with sociostructural factors (impediments/barriers and facilitators), managing outcome expectations, and setting goals to improve health behaviors.[7] The Behavior Change Techniques (BTC's) used to promote health behavior change are defined according to the BCT Taxonomy version v1 [8] and listed in supplementary table A. Extended information on the content per dietary counselling session is provided in supplementary table B.

Supplementary table A: Overview of Behavior Change Techniques (BCT's) used to promote health behavior change in the PADOVA dietary intervention

Theory	Construct	BCT[8]	Description of BCT
Social Cognitive Theory	Self-efficacy	Graded tasks	To promote self-efficacy, the dietitian will stimulate the participant to set easy to perform and achievable individual goals, and will promote gradually making individual goals more difficult until the recommendation is met.
	Outcome expectations	Comparative imagining of future outcomes	The dietitian will prompt or advise the imagining and comparing of future outcomes of changed versus unchanged behavior.
		Vicarious consequences	The dietitian will prompt observation of the consequences (including rewards and punishments) for others when they perform the behavior.
	Goal setting	Goal setting (outcomes of) behavior	Individual goals with regard to dietary intake/weight will be set by the participant in consultation with the dietitian.
	Sociostructural factors	Problem solving	The dietitian and patient discuss factors that could influence achieving each goal, as well as strategies to overcome possible barriers and/or strategies to increase facilitators to achieving each goal.
		Social support (practical)	The dietitian gives advice on finding social support (e.g. practical help from family or friends) for behavior change in order to reach individual goals.
		Habit formation	The dietitian will advise on rehearsal and repetition of the behavior in the same context repeatedly so that the context elicits the behavior.
		Avoidance/reducing exposure to cues for the behavior	The dietitian will advise on how to avoid exposure to specific social and contextual/physical cues for the behavior, including changing daily/weekly routines.
		Restructuring the physical environment	The dietitian will facilitate change or advise to change the physical environment in order to facilitate performance of the wanted behavior or create barriers to the unwanted behavior.
		Restructuring the social environment	The dietitian will facilitate change or advise to change the social environment in order to facilitate performance of the wanted behavior or create barriers to the unwanted behavior.
Information about antecedents		The dietitian will provide information about antecedents (social, environmental situations or events, emotions, cognitions) that reliably predict performance of the behavior.	
Self-reward		The dietitian will prompt self-praise or self-reward if and only if there has been effort and/or progress in performing the behavior.	
Reduce negative emotions		The dietitian will advise on ways of reducing negative emotions to facilitate performance of the behavior (includes stress-management).	
Motivational interviewing		Pros and cons	The dietitian will advise to identify and compare reasons for wanting (pros) and not wanting (cons) to

		change the behavior (includes decisional balance).
	Comparative imagining of future outcomes	The dietitian will prompt or advise the imagining and comparing of future outcomes of changed versus unchanged behavior.
	Social support (unspecified)	The dietitian will advise on and how to arrange social support (e.g., from friends, family, buddies) or non-contingent praise or reward for performance of the behavior. Includes encouragement and counselling when directed at the behavior.
Other		
	Credible source	Dietitian from hospital provides counselling.
	Feedback on behavior	The participant will receive feedback from the dietitian on diet quality, and on the extent to which they meet the World Cancer Research Fund (WCRF)/American Institute for Cancer Research (AICR) [9] recommendations and the protein-goals [10,11].
	Feedback on outcomes behavior	The participant will receive feedback from the dietitian on their weight, Body Mass Index, body composition.
	Information about health consequences	The dietitian will inform the participant about the influence of lifestyle-related factors on the occurrence of cancer and about the potential positive effects of increased physical activity and a healthy diet (including the effect of habitual protein consumption during exercise) throughout chemotherapy.
	Instruction on how to perform the behavior	The participant will receive information from the dietitian on the WCRF/AICR recommendations (leaflet).
	Adding objects to the environment; self-monitoring of behavior; self-monitoring of outcomes of behavior	The participant will receive a self-monitoring log from the dietitian in which they can log their weight and diet. They are encouraged to weekly log their weight, and to daily log their dietary intake, with flexibility to meet individual needs and preferences.
	Action planning	An action plan for each individual goal will be discussed by the participant and dietitian.
	Discrepancy between current behavior and goal	The dietitian will point out potential discrepancies between patients' current behavior and each goal during each subsequent session.
	Review behavior goals/review outcome goals	The self-monitoring logs will be discussed with the oncology dietitian during every counselling visit to be able to monitor progress. Each goal will be reviewed and may be modified if necessary. Also, new goals may be set.
	Social reward (positive reinforcement)	The dietitian will congratulate the patient in case of success.
	Verbal persuasion about capability	The dietitian will tell the person that they can successfully perform the wanted behavior, arguing

against self-doubts and asserting that they can and will succeed.

Supplementary table B: Overview of the content of the PADOVA dietary counselling sessions

Counselling session	Content
First counselling session	 - Introduction of dietitian and aim of dietary counselling sessions - Anthropometric measures  o Current weight and weight history o Height o Body Mass Index o Body composition - Dietary assessment  o Nutrition-related illnesses or symptoms (e.g. reduced appetite, nausea, vomiting, gastrointestinal problems, chewing or swallowing difficulties) o Relevant social factors (e.g. social support) o Dietary analyses of current nutritional intake o Current exercise and physical activity level - Assessment of energy [12,13] and protein [10,11] requirements using multiple formulas - Dietetic diagnosis (synthesized information from anthropometric measures and dietary assessment) - Provide feedback on patients' weight, body composition and dietary intake - Providing information about the influence of lifestyle and body weight related factors on the occurrence of cancer and about the potential positive effects of increased physical activity and a healthy diet (including the effects of habitual protein consumption during exercise) throughout chemotherapy. - Set individual goals and action plans to achieve goals (depending on current nutritional status) - Discussion of factors that could influence achieving each goal, as well as strategies to overcome possible barriers and/or strategies to increase facilitators - Hand out self-monitoring logs in which patients can log their weight, dietary intake and/or physical activity.
Second – fifth counselling session	 - Anthropometric measures  o Current weight and Body Mass Index o Body composition (every other counselling session) - Dietary assessment (if changed) - Assessment of energy and protein requirements (if changed) - (Revision of) dietetic diagnosis - Discussion of filled in self-monitoring logs - Discussion of potential discrepancies between current behavior and each goal - Review and if necessary modification of goals and action plans
Last counselling session	 - Same content as second to fifth counselling session - Discussion and encouragement of self-regulation strategies to be able to maintain adherence to the World Cancer Research Fund/American Institute for Cancer Research recommendations after the end of the intervention [9]

2. A more detailed description of the recruitment procedures should be provided. Explicitly addressing the process of how the clinical referrals are obtained would be instructive.

Author response:

We thank the reviewer for the suggestion to add a more detailed description of the recruitment procedure. We have added a more detailed description of the recruitment procedure to the manuscript (lines 138 – 140): 'After diagnosis and before the start of neo adjuvant chemotherapy (\pm 2 weeks) or adjuvant chemotherapy (\pm 4 weeks), the gynaecological oncologist informs patients about the PADOVA study during an out-patient clinic visit.'

3. A more detailed description of the progressions implemented in the exercise prescription is also warranted. Specifically, the description of how load, volume, and volume-load progressions of the resistance exercise component are implemented is underdeveloped and further detail of this procedure would be particularly informative.

Author response:

We agree with the reviewer that it is important to describe in more detail how load, volume, and volume-load progressions of the resistance exercise component are implemented. We have incorporated details on this procedure concurrently with the details about the exercise training principles in lines 219-235. Please see our response to review comment 1.

4. Description of the specific strategies to promote adoption and maintenance of change in dietary intake are not sufficiently developed and could be more clearly articulated.

Author response:

We agree with the reviewer that it is important to describe the specific strategies to promote adoption and maintenance of behavior change. Please see our response to review comment 1 for a more detailed description.

5. The 15% drop out rate and anticipated effect size for body composition ($d = .55$) both appear overly optimistic and should be more explicitly justified.

Author response:

We would like to thank the reviewer for pointing out the need to further elaborate on the justification of the drop-out rate and anticipated effect size we used in our power calculation.

The dropout rate of 15% was based on observed drop-out rates in our previous exercise trials in patients with cancer during chemotherapy treatment (10%),^[14] in a group of patients with cancer after completion of chemotherapy treatment (10%),^[15] and in patients with multiple myeloma or (non) Hodgkin lymphoma who received high dose chemotherapy and autologous stem cell transplantation (15%).^[16] Based on the reported drop-out rates from our previous studies, we consider the 15% drop out rate to be a reasonable estimate. The drop-out rate in the PADOVA study is 12% thus far. To justify the drop-out rate used in our power calculation, we have added the following additional information in lines 335-336 'Dropout rates are based on the 10-15% dropout rates from previously conducted Dutch exercise trials in patients with cancer.'^[14-16]

The anticipated effect size for body composition is based on information on both body fat and muscle mass obtained from previous studies available at that time.^[17-21]

Sample size calculations were based on a relevant and feasible difference of 3% in muscle mass and percentage body fat, and a standard deviation of 5%. To obtain a power of 80% and a two-sided alpha of 5%, 122 were needed. From observational studies among patients with breast cancer treated with curative intent, it is known that adjuvant chemotherapy is associated with a 1-4% increase in percent body fat.^[17,18] Additionally, previous studies examining lifestyle interventions showed changes in body fat between 2.5 and 5.3%^[19,21] with a SD of 2.7 and 4.1 respectively. A 3% difference in muscle mass was considered relevant based on the findings that patients lose on average 6.1% (corresponding to 1.7 kg in men and 1.1 kg in women) of muscle mass during palliative chemotherapy treatment, and that patients with losses of 9% had significantly higher mortality risk.^[20] To further

clarify the sample size calculation in the manuscript, we have added the following information (in bold) in lines 327-334 'Sample size calculation is based on the results of previous RCT among patients with breast cancer that evaluated the effects of a combined aerobic and resistance exercise intervention (similar to PADOVA study),[14] a pilot study among patients with ovarian cancer that evaluated the feasibility of an exercise intervention[22] and on clinically relevant differences in body composition and peak oxygen uptake.[17-23]. With 53 patients per study-arm, we are able to detect a clinically relevant between group difference in effects directly post-intervention on physical function (10 point), physical fatigue (2.7 points), body composition (3% in percentage of body fat and muscle mass) and 10-15% difference in peak oxygen uptake ($\alpha=0.05$; power= 0.80).

6. From a conceptual perspective, the intervention appears to involve considerable contact with specialists with expertise in exercise physiology, physical therapy, and dietetics. I urge the authors to explicitly address the implications of this approach for the reach, scalability and overall impact of the proposed intervention.

Author response:

The intervention is delivered by physical therapists from first line and dietitians from second line health care. The physical therapists are specifically educated to provide supervised exercise to patients with cancer and are affiliated with a nation-wide oncology network which include > 600 physical therapy practices across the Netherlands.

The oncology dietitians are employed in the hospitals from which the patients are recruited. When patients are not able to visit the hospital (e.g. due to long travel distances), the dietary intervention can be (partly) provided by telephone.

The close collaboration with these networks for physical therapists and dietitians facilitates reach and scalability of the PADOVA intervention. Recently, a Dutch network for dietitians specialized in care of oncology patients was founded. This will allow future patient referral to a dietitian close to patients' home, further improving reach and scalability of the PADOVA dietary intervention.

We have added the following text to method and discussion to address the implications on reach, scalability and overall impact of the PADOVA intervention:

Method lines 212-215: 'The exercise sessions are supervised by a physical therapist specifically trained in treating oncology patients, to inform patients on and monitor appropriate and safe exercise strategies. These physical therapists are affiliated with a nation-wide network that includes >600 physical therapy practices. This enables to offer the intervention close to a patients' home.'

Discussion lines 371 – 375: 'Close collaboration with physical therapists specifically educated to supervise patients with cancer, affiliated with a network throughout the Netherlands has the advantage that the exercise intervention can be offered close to a patients' home. This increases reach and scalability of the intervention. Consequentially, this might also lead to small variations in the implementation of the intervention protocol by each individual physical therapist.'

Reviewer: 2

General comments

The current study reports on the study protocol for a randomized controlled trial of physical activity and dietary intervention for women with ovarian cancer undergoing chemotherapy. The study will test and compare the effects of a tailored exercise and dietary intervention on primary outcomes of body composition, physical function, and fatigue. The study will also examine and compare the effects of these interventions on a battery of secondary outcomes assessed before starting chemotherapy, 3 weeks after completing chemotherapy and 12 weeks later. The aims of the manuscript include describing the tailoring of PA and Diet to the comorbidity status and presence of adverse events (AE's) in patients. The proposed study enrolled its first patient in February 2018 and is currently open for patient inclusion targeting a total of 122 patients by December 2020. The target population is epithelial ovarian cancer patients scheduled to receive (neo) adjuvant chemotherapy.

The interventions are designed to be 2 x 1-hour supervised sessions per week, including both

resistance and aerobic exercise. Dietary counseling includes individually tailored 30-minute sessions every 3 weeks during chemotherapy. Waitlisted control participants will receive education and information about diet and activity with access to counseling sessions upon request following the first assessment.

The study is relatively novel in that it will examine the effectiveness of a tailored combination lifestyle intervention (shown to be effective in some studies) in an understudied cancer population undergoing chemotherapy. A secondary aim of the study is to evaluate the intervention processes. I commend the authors for preparing this study and for the detail included in the reporting of the protocol.

Strengths of the study are the randomized design, the use of extensive objective measures of outcomes, detailed process on tailoring the intervention (i3-s) and process evaluation. A limitation of the design is the lack of long term follow up due to the control group being provided with access to a limited number of intervention sessions.

Thank you for the opportunity to review your work and I wish you luck with the study. I have only a limited number of queries.

Author response:

Thank you for your compliments and for investing your time to review our manuscript.

Methods and analysis

1. Under the description of i3-S methodology, on page 9, lines 187-196, the authors state that previous papers have discussed comorbidities and AE's and that they will therefore not be included in this paper. Just to clarify—you will still refer to these guidelines (previously published) when determining tailoring considerations for your ovarian population, you are just not writing about them in this paper?

Author response:

Indeed, we have developed a protocol for physical therapists with determining tailoring considerations including all comorbidities and AE's. However, due to the restricted word count and to prevent publication duplicates, we have chosen to refer to the publications describing to specific comorbidities and AE's tailored interventions, and to only list additional comorbidities and AE's that have not been discussed previously.

To clarify this, we have now made the following changes (in bold) to the manuscript (lines 190-197): 'The following comorbidities (i.e. hypertensive diseases, ischaemic heart diseases, other forms of heart disease, chronic lower respiratory diseases, Diabetes Mellitus), and adverse effects (clinical parameters such as leukopenia/neutropenia, trombopenia, anemia; and symptoms of dyspnea, nausea, vomiting or diarrhea, skin and nail changes, fever, dizziness, decreased or increased heart rate, change in body weight, depression, numbness/loss of sensation, hearing and/or visual impairments, fatigue, pain and chest pain) were described in previous publications[24,25] or nutritional guidelines[11,26-28] and are not presented in this paper.

2. Under the Exercise Intervention description, lines 226-228, the authors describe the recommendation to include physical activity independently at least 3 additional days per week. Given the listed (table 1) comorbidities in ovarian cancer patients, how do the researchers plan to monitor the safety of patients in the home or unsupervised-setting? Specifically as regards to potential cardiovascular complications.

Author response:

We understand the concern of the reviewer on safety of patients with regard to potential cardiovascular complications. Exercise clearance is given prior to participation in the PADOVA study by the treating gynaecological oncologist and at the start of the exercise intervention by means of medical screening (Physical Activity Readiness Questionnaire [29] and a maximum exercise test on a cycle ergometer. If needed, exercise recommendations or execution of the maximum exercise test is adjusted in consultation with a cardiologist. The physical therapists supervising the intervention are

specifically educated to monitor safety of exercise, and will inform patients on appropriate and safe exercise strategies.

To clarify this in the manuscript, we have added information (in bold) in lines 212-214: 'The exercise sessions are supervised by a physical therapist specifically trained in treating oncology patients, to inform patients on and monitor appropriate and safe exercise strategies.'

3. Can the authors provide, from their pilot study, information about the typical time between diagnosis (histological confirmation of ovarian cancer) and decision to treat with chemotherapy? While this may be reported in the feasibility paper you cite, and vary across settings/countries, it is important to consider the feasibility of enrolling patients to study during this demanding window. Any information about expected timeframes would be beneficial to readers and researchers.

Author response:

Between histological confirmation of ovarian cancer and the decision for chemotherapy treatment is a time window of approximately 2 weeks in patients treated with neo adjuvant chemotherapy and 4 weeks for patients treated with adjuvant chemotherapy. We have included information on this timeframe in lines 138 – 140 of the manuscript: 'After diagnosis and before the start of neo adjuvant chemotherapy (± 2 weeks) or adjuvant chemotherapy (± 4 weeks), the gynaecological oncologist informs patients about the PADOVA study during an out-patient clinic visit'.

Reviewer: 3

This study describes an evaluation of a physical activity and dietary intervention for women with ovarian cancer. This is an important topic as although the benefits of such interventions have been demonstrated in other areas of cancer, primarily breast cancer, there has been little work in ovarian cancer. The paper is mainly well written, although a proof read to ensure correct grammar and English usage would be advised – in particular use of tense. As this is written as a protocol paper the future tense will most commonly be needed.

Author response:

Thank you for your time and effort invested in reviewing our manuscript. We have carefully read the manuscript on the use of the English language, and consistent use of future tense, and have made adjustments in track changes in the manuscript accordingly.

There are a number of recommendations which I would encourage the authors to consider:

Background

1. I find the presentation of the dietary data somewhat confusing and would encourage the authors to review the presentation of this information. For example it is suggested that a high percentage of individuals are overweight or obese, but yet malnutrition is also common. Is there any overlap between these groups or is there an additional underweight group. Weight lost is suggested to be associated with lower survival rate but is this true regardless of baseline BMI or are there benefits in say the overweight group from losing weight but risks for an originally underweight group. Exercise is suggested to be helpful in preventing weight gain but the above is not suggestive that this is a common problem with ovarian cancer.

Author response:

We understand that the dietary information as presented in the introduction of the manuscript may be somewhat confusing as there are different components presented: malnutrition, overweight/obesity, and sarcopenia. These components may be present separately or combined, e.g. patients can have a high BMI but still be malnourished. Independent from Body Mass Index (BMI), half of the patients with ovarian cancer are at risk of malnutrition before start of or during medical treatment.[30,31] There could be overlap between BMI and malnutrition, however this is not clear from observational studies as they do not compare body weight/BMI and (risk of) malnutrition.

We have added the following text to clarify presentation of dietary data in the introduction (lines 81-85): 'Additionally, previous studies reported that half of the patients suffer from sarcopenia (i.e. loss of skeletal muscle mass) or malnutrition at diagnosis and that the prevalence increased during

neoadjuvant chemotherapy.[30-33] Independent from the presence of sarcopenia and malnutrition, studies reported that 24-57% of patients with ovarian cancer are overweight and 10-35% are obese.[32-38]

Methods

2. My key concern with the study is the potential for contamination or lack of difference in intervention received between the groups. Both groups receive a brochure on diet, exercise and weight recommendations and the control group get an intervention after T1, the significant limitation of this should be made clearer and suggests more of a feasibility design than large scale RCT as implied in the introduction.

Author response:

The brochure on exercise, diet and weight recommendations contains general information on physical activity, diet and body weight recommendations for cancer survivors [39]. These recommendations are not tailored to the individual patient. Additionally, the patients in the control group will receive usual care during chemotherapy treatment that does not include structured and/or supervised exercise and/or dietary intervention. As general information on health behaviour is generally not sufficient to change behaviour, we expect contamination to be limited. Contamination is further reduced by offering exercise and dietary counselling sessions to the control group after completion of chemotherapy.

We have now clarified this in the manuscript (in bold) in lines 147-149: 'After randomisation, patients in both the intervention and the control group receive a brochure with general information on physical activity, diet and body weight recommendations for cancer survivors [39]. These recommendations are not individualized, nor supervised.'

'In order to prevent nonparticipation and drop-out in the control group, participants in the control group are offered a maximum of three exercise and three dietary counselling sessions in twelve weeks after completion of chemotherapy and the first follow up measurement. (lines 266-268).

3. More detail should be provided on the development of the intervention and description of the intervention, including possible theory and logic model if relevant (see below). Was PPI included in intervention development?

Author response:

We thank the reviewer for raising the suggestion to describe the development of the intervention in more detail. We have now elaborated on the exercise training principles (lines 218-219), the use of Motivational Interviewing techniques (lines 241-243), the Social Cognitive Theory (lines 243-249) and associated Behavior Change Techniques (listed in supplementary file C). Please see our response to review comment 1 of reviewer 1 for a more elaborate description of the development of the intervention, including the theoretical basis of the intervention.

PPI was involved in the development of study specific patient information and participants will be interviewed on the acceptability of the intervention. This information is important for implementation of the intervention in clinical practice. Please find a description of PPI involvement In lines 382-387 of the manuscript. We have now explicitly mentioned that patients were interviewed on acceptability of the intervention (see changes in bold in lines 384-385): 'During the study patients will be interviewed (e.g. on acceptability of the intervention), as part of the process evaluation, to improve the intervention.'

4. Whilst it is important that co-morbidities were considered and accounted for in the exercise intervention I do not feel the level of detail that is provided here, including 2 detailed tables is of particular interest. I would have thought a couple of lines explaining the strategy and a brief table

showing common adjustments would be sufficient in the main text with more detailed discussion including tables 1 and 2 in supplementary information if the authors feel this is really necessary.

Author response:

We have taken the suggestion of the reviewer to include tables 1 and 2 in supplementary information instead of in the manuscript into consideration. However, we think it is important to include in depth information on the development of the intervention using the I3-S method provided in tables 1 and 2 in the manuscript. Since one of the aims of this study was to describe how the exercise and dietary intervention were tailored via the I3-s strategy to ovarian cancer specific comorbidities, disease- and treatment induced adverse effects to optimize intervention feasibility and study retention rates. Most previous studies examining exercise or dietary interventions in patients with cancer were conducted in patients with breast cancer. It is unclear whether the effects of exercise and dietary interventions found in patients with breast cancer can be generalized to women with ovarian cancer. Compared with breast cancer, ovarian cancer is often detected in a more advanced stage [40] and in older women. Ovarian cancer also has a substantially different treatment trajectory, i.e. different type of chemotherapy and other adjuvant therapy regimens (lines 98-103). To optimize intervention feasibility and study retention rates, we aim to offer exercise and dietary intervention that are specifically tailored to the comorbidities, disease- and treatment induced adverse effects that individual patients with ovarian cancer may face (lines 115-117).

5. In contrast there appears to be very little detail on the exercise and dietary interventions including how these were delivered and how they were developed. For example it is important to know for the exercise intervention what the training of the physical therapist was, was exercised delivered in groups, if so what size were groups were these closed or open, were there any behavioural elements to promote exercise. Also how long was exercise delivered for – throughout chemotherapy? In which case was this a standard period or was it variable

Author response:

We thank the reviewer for the suggestion to further elaborate on the development and the content of the intervention. We have added additional information (in bold) on the training of the physical therapist and group sizes and have made small adaptations accordingly in lines 212-217 of the manuscript. 'The exercise sessions are supervised by a physical therapist specifically trained in treating oncology patients, to inform patients on and monitor appropriate and safe exercise strategies. These physical therapists are affiliated with a nation-wide network that includes >600 physical therapy practices. This enables to offer the intervention close to a patients' home. Depending on the physical therapy practice, patients train in small groups with other (non-)PADOVA patients (with cancer).'

In lines 158 – 160 we describe that the intervention starts at the first cycle of chemotherapy and continues until three weeks after the last cycle. We have added information to clarify the duration of the intervention: 'In general, patients receive six cycles of chemotherapy (duration of ± 18 weeks). In case of dose delay or discontinuation of chemotherapy treatment, duration of the intervention and range of time between study measurements might differ between patients.'

Additional information on behavioral elements of the intervention are described under review comment one of reviewer one.

6. For the dietary intervention what motivational interviewing techniques were used, what was the training of the dietitians in these therapies

Author response:

Dietitians are trained in motivational interviewing techniques as this is incorporated in the Bachelor of Science, Nutrition and Dietetics.

We have included information on the motivational interviewing techniques used in the dietary intervention in lines 241-243: 'Motivational interviewing is an effective counselling method for achieving health behavior change using techniques as reflective listening and summarizing, focusing on what the patient wants, thinks and feels).[5] Also see table 1 (review point 1 of reviewer 1) with

behavior change techniques (BCT's) for an overview for the BCT's within the context of motivational interviewing that are applied in the PADOVA intervention.

7. Control group – it is noted that individuals did not receive structured exercise or dietary counseling, however if they were malnourished at baseline would this not be addressed in standard care?

Author response:

We thank the reviewer for addressing standard care in patients who are malnourished. As part of standard care for patients with ovarian cancer in the Netherlands, patients are referred to a dietitian when malnutrition is detected by their treating gynaecological oncologist.

We have added information on the assessment of malnutrition in lines 264 – 265: 'Women in the control group will receive usual care during chemotherapy, which includes referral to a dietitian when malnutrition is detected by the gynaecological oncologist.'

8. It is not clear what the range of times T1 could be collected – are there different periods chemotherapy is given for. Is T2, 12 weeks after t1, or baseline?

Author response:

'Baseline measurements are conducted before randomization and the start of chemotherapy (T0), the second measurement three weeks after completion of chemotherapy (T1) and the last measurement (T2) is conducted twelve weeks after T1. In general, patients receive six 3-week cycles of chemotherapy treatment. However, in some cases, for example due to toxicity, dose delay or discontinuation may occur, influencing the time schedule of chemotherapy treatment. To clarify the range of time between the baseline- (T0), first follow up- (T1) and last measurement (T2) we provided additional information in bold in the manuscript.

Lines 158 – 160: 'In general, patients receive six cycles of chemotherapy (duration of ± 18 weeks). In case of dose delay or discontinuation of chemotherapy treatment, duration of the intervention and range of time between study measurements might differ between patients.' Lines 272-274: 'Baseline measurements are conducted before randomization and the start of chemotherapy (T0), the second measurement three weeks after completion of chemotherapy (T1) and the last measurement (T2) twelve weeks after T1.'

9. I am unclear what the purpose of the blood sample for a biobank is. Was consent for this a requirement for consent to the main study. A statement on ethical approval for this as well as other aspects of the study is important.

Author response:

The PADOVA study, including blood sample collection for the PADOVA biobank, is approved by the medical ethical committee of Amsterdam UMC (lines 122 – 124). We agree with the reviewer that we could describe blood sample collection for the PADOVA biobank in more detail in the manuscript. We have added more detailed information in lines 296-298: 'During the visit on T0 and T1, a venous blood sample is drawn and stored in the PADOVA biobank for future biomarker studies (e.g. to assess immune system functioning). Because collection of blood samples is an addition to participation in the PADOVA study, an additional informed consent is obtained.'

Process Evaluation

10. It is encouraging to see a process evaluation included, however this is not completely congruent with the intervention description which provides little consideration of psychosocial factors. If, and if so how, these factors were targeted in the intervention should be made clear, including a logic model and description of underlying theory. If these factors were not targeted by the intervention a rationale for why these are being evaluated as mediator variables would be helpful. In contrast if the variables are not targeted by the intervention but are being considered as predictor variables this should be made clear and moderator analysis considered rather than mediator analysis.

Author response:

We agree with the reviewer a detailed description of the underlying theory for the PADOVA intervention should be provided. Behavioral determinants, including social cognitive theory constructs are targeted within the PADOVA intervention to achieve health behavior changes. See review comment 1 of reviewer 1 for more detailed information on the theoretical basis of the PADOVA intervention. An overview of Behavior Change Techniques (BCT's) used in the PADOVA intervention is provided in supplementary file C and/or review comment one of reviewer one.

To examine whether intervention effects on body composition, physical function and fatigue are mediated by changes in physical activity and fitness and/or dietary intake (e.g. protein intake), a series of regression analysis according to the product-of-coefficients test will be conducted (lines 344-346).'

11. There is a statement about fidelity being measured, again this is encouraging however fidelity of what should be stated as should how it will be measured and analysed.

Author response:

Fidelity is defined as the extent to which the intervention was delivered as described in the intervention protocol. Fidelity is assessed via a PADOVA intervention checklist filled in by physical therapists and dietitians, using the following statements: Was the intervention delivered as described in the protocol? (answer categories: yes/no). In case the intervention is not delivered according to protocol, physical therapists or dietitians are asked to state the number of times they made adjustments to the protocol, to describe these adaptations, and to describe the reason for the adjustments to the protocol. We have added a brief description of fidelity and how this is measured in table 4 of the manuscript: 'Fidelity: extend to which the intervention was executed as prescribed in the protocol (e.g. reasons for and amount of adaptations to the protocol).'

12. it is stated that interviews will be analyzed by a data analysis program. Which program will this be? Do the authors mean a program will be used to assist the data analysis process rather than program actually do the analysis?

Author response:

Data from interviews is analyzed using Atlas.ti. We have adjusted information (in bold) on this qualitative data analysis program in the manuscript in lines 315-316: 'Interviews will be transcribed verbatim, coded in several phases[41,42] and analyzed with help of Atlas.ti.'

Discussion

13. I feel it is overselling the study to call it a large RCT, the control group will be contaminated post intervention so no medium or long term outcomes can be examined. I appreciate this is recognized as a limitation however this presentation should be more balanced.

Author response:

As only small RCT's have been conducted in patients with ovarian cancer and ovarian cancer is a relatively rare type of cancer, we consider this as a relatively large RCT. We agree with the reviewer that this is relative and therefore removed the term 'large' from the manuscript.

We also agree with the reviewer that it is a limitation of the study that we cannot examine medium or long term outcomes, which was mentioned as such in the discussion of the manuscript. We have elaborated on the choice of study design in lines 266-268. Please also see our response to review point 2.

14. Also although the study calls itself multcentre only one centre is named for recruitment and it appears a single physical therapist is used to deliver the intervention. Being more reflective on this and the implications of what would be needed for full implementation (including training programmes) if successful would be helpful.

Author response:

Thank you for pointing out that our presentation of the recruiting hospitals was not fully clear. Patients are recruited from two regional gynaecological oncology centers, which include all collaborating

peripheral hospitals. We have also added multiple centers (in bold) to the manuscript which started recruitment for the PADOVA study in December 2019. We have clarified this in lines 135-138 of the manuscript: 'Patients are recruited from the Center of Gynaecologic Oncology Amsterdam (which is a collaboration of all gynaecological oncologists of Amsterdam UMC, the Netherlands Cancer Institute – Antoni van Leeuwenhoek and affiliated peripheral hospitals) and Catharina hospital and its collaborating peripheral hospitals in the South of the Netherlands.'

The dietary intervention is provided by dietitians in the hospital the patient is recruited. The exercise intervention is supervised by a physical therapist close to a patients' home. For this collaboration we use a network of physical therapists specialized in the care of patients with cancer. This network now includes >600 physical therapy clinics throughout the Netherlands. A more detailed description of reach and scalability of the intervention is presented at review comment six of reviewer two.

VERSION 2 – REVIEW

REVIEWER	Brian C. Focht, PhD, FACSM, CSCS The Ohio State University, USA
REVIEW RETURNED	09-Jun-2020

GENERAL COMMENTS	The authors have completed a comprehensive revision that addresses all the primary concerns raised in the initial review in a satisfactory manner. I have no further proposed revisions to the protocol ms.
---

REVIEWER	Alexander R. Lucas Virginia Commonwealth University School of Medicine, United States
REVIEW RETURNED	29-Jun-2020

GENERAL COMMENTS	I thank the authors for their detailed responses to all concerns raised in the initial review. I feel they have addressed mine and other reviewers' requests and have no further suggestions or required revisions.
---

REVIEWER	Liz Steed Queen Mary University of London, UK
REVIEW RETURNED	12-Jun-2020

GENERAL COMMENTS	I am satisfied that this paper has addressed all my concerns fully and happy to suggest acceptance for publication
--